# Facilitating implementation of an evidence-based method to assess the mental health of 3–5-year-old children at Child Health Clinics: A mixed-methods process evaluation

**Elisabet Fält** *, **Raziye Salari, Helena Fabian, Anna Sarkadi**

Department of Public Health and Caring Sciences, Child Health and Parenting (CHAP), Uppsala University, Uppsala, Sweden

* Elisabet.Falt@pubcare.uu.se

## Abstract

### Background

A number of instruments for identifying mental health problems in children are available, but there is limited knowledge about how to successfully implement their use in routine practice. The Strengths and Difficulties Questionnaire (SDQ) is an instrument with sound psychometric properties. Because using multi-informant SDQs when assessing young children has been emphasized, parent- and preschool teacher reports on the SDQ were introduced at Child Health Clinics in a Swedish municipality. This paper aimed to describe a facilitation programme developed to support the introduction of SDQ in clinical practice and evaluate how nurses perceived the facilitation strategies used. Moreover, the dose (delivery) and reach (response rate and population coverage) of the questionnaires were assessed.

### Methods

The mixed-methods process evaluation was guided by Moore et al.'s framework. Process data were excerpted from monitoring data, the trial database, research group documents, study materials, group interviews with nurses, and a survey on nurses' opinions and experiences of the screening method and the implementation process. Data were analysed using descriptive statistics and qualitative content analysis.

### Results

Facilitation strategies used included: educational meetings, educational outreach visits, newsletters, facilitative administrative support, and adaptations made in procedures and materials when required. Although nurses described a variety of barriers at the organisational and individual level, they were in favour of using the SDQ in clinical practice and emphasised the importance of the facilitation strategies used for its implementation. While dose levels (77–91%) indicated that nurses essentially delivered the intervention as intended, parental response rates remained between 54 and 63% and population coverage at around 50%, throughout the intervention period.

**Data Availability Statement:** Ethical clearance was granted by the Regional Ethical Review Board in

Uppsala, Sweden (Dnr 2012/437). The participants in this study have not consented to deposition of the data. Data also contain sensitive information on children. Due to ethical restrictions related to protecting patient confidentiality, all relevant data are available upon request and approval from the Senior Registrar Clerk at Uppsala University. Interested researchers may contact the Principal Investigator, Professor Anna Sarkadi (anna. sarkadi@pubcare.uu.se) or Uppsala University (registrator@uu.se), to request the data used for the analyses in this paper.

**Funding:** Funding was received from the common grant of major Swedish research funders (FORMAS, Vetenskapsrådet, FAS and VINNOVA), termed "Mental health of children and adolescents" (grant number 259-2012-68) and from Uppsala County Council's fund for research. The funders had no role in study design, data collection and analysis, decision to publish, or preparation of the manuscript.

**Competing interests:** The authors have declared that no competing interests exist.

## Conclusion

The facilitation program was perceived to support the implementation of the SDQ at the yearly check-ups in the child healthcare setting, but further efforts are required to reach all families.

## Introduction

### Detection of mental health problems in the Swedish Child Health Services

In Sweden, Child Healthcare Services (CHS) are responsible for offering a universal programme, including routine health check-ups, without charge, to all parents with children aged 5 and under. Given a reach of more than 95% of the 0–5-year-old population [1], CHS could play a pivotal role in identifying mental health problems in children. However, structured methods are not used for this purpose. Also, while a multiple informant approach is considered best practice for the assessment of children's mental health [2] and preschool teachers have been recognised as important informants in identifying children with mental health problems [3], current assessments at Child Health Clinics (CHCs) rely mainly on parents' description of their child's functioning and health. Hence, the current capacity of CHS to effectively monitor mental health problems is limited. The introduction of a structured tool to assess children's emotional and behavioural problems through parent and teacher reports could reduce the risk of mental health problems being left undetected.

Previously, we have explored nurses, preschool teachers and parents' perspectives on using the Strengths and Difficulties Questionnaire (SDQ), a screening tool for behavioural and emotional problems, in the child healthcare setting [4]. While these first results were encouraging, they also provided us with a deeper insight of the complexity of the intervention, informing us about ongoing facilitation needs.

Although implementing evidence into practice is challenging, facilitation has been suggested to affect the use of evidence [5]. A range of facilitation strategies have therefore been designed and used in the current trial to support implementation.

### Evaluation of implementation strategies

There are many challenges inherent in implementation efforts [6], and interventions within healthcare are often *complex*, presenting several challenges for evaluators [7]. Process evaluation efforts are therefore important to increase our understanding of factors associated with complex intervention successes and failures [8]. In the past ten years, process evaluation, as such, has received considerable attention, through the Medical Research Council (MRC) guidelines [9, 10], practical recommendations [11] and finally, a framework for process evaluation developed by Moore et al. [12]. This framework [12] argues for a structured approach to conducting a process evaluation, and recommends that such evaluation should include investigation of three key components: *context*, *implementation* and *mechanisms of impact*. The framework describes the components and provides a model visualizing these three functions of a process evaluation, and the relation among them. The model provides a basis for assessing and understanding factors promoting or inhibiting the embedding of complex interventions in everyday health care practice, and informs interpretation of outcomes. In this paper, we use the framework [12] to understand and evaluate the complex process involved in facilitating the implementation of a new method at Child Health Clinics.

## Aim

This process evaluation aimed to: 1) Describe the facilitation programme developed to support the introduction of SDQ in clinical practice, 2) Quantify dose and reach of the intervention and describe change over time and 3) Explore nurses' perception of the facilitation strategies and the intervention.

## Methods

### Design

This study used a prospective, mixed-methods design wherein qualitative and quantitative data were given equal emphasis. The guidance for process evaluation as suggested by Moore et al. [12] was used to structure the data for the facilitation process: *context*, *implementation* and *mechanisms of impact*. In this paper, we focus on the *implementation* and the *mechanisms of impact* aspects. The *context*, in terms of barriers and enablers to implementation, has been reported previously [4].

**The facilitation program.** Facilitation has been suggested to affect the uptake of evidence into practice [5]; therefore, a facilitation programme including three empirically justified strategies was developed to support the implementation process. The facilitation strategies focused on addressing issues relating to fidelity i.e. enabling nurses to deliver the intervention according to plan, and in the spirit in which it was intended. Facilitation strategies included:

1. Having regular educational outreach visits with nurses at the local CHCs,

2. Having educational meetings with all nurses involved in the trial,

3. Sending out newsletters to CHCs.

### Causal assumptions

The intention of the facilitation programme was to inform the nurses about the purpose and benefits of the intervention, and to provide them with supportive supervision and feedback on their performance. The facilitation programme was assumed to trigger behavioural change by increasing nurses' beliefs in and ownership of the intervention, as well as promoting their self-efficacy to assess children using SDQ and being inclusive of all families.

### Setting

This process evaluation was undertaken alongside the Children and Parents in Focus trial, a large-scale randomised controlled trial (RCT), which ran between September 2013 and October 2017. The RCT was designed to evaluate the effectiveness and cost effectiveness of a universally offered parenting programme in the Swedish context [13]. Data collection of the outcome measures (including SDQ) was conducted through the regular child health services in Uppsala municipality (population 225,164). Of the 21 CHCs in the municipality, nineteen participated in the first year of the trial. In subsequent years, the number of participating CHCs varied (range 18–19); specifically, two CHCs joined the study and two CHCs were closed at different timepoints during the trial. All parents of 3, 4 and 5-year-old children enrolled at the CHCs were invited to participate in the Children and Parents in Focus trial. The total number of eligible children was estimated to be 6,882 in the first year of the study and 7,056, 7,316 and 7,689 in the following years. Approximately 160 preschools in the municipality (all except 3) participated.

**The intervention–information sharing using the SDQ.** Because collection of the SDQ was not routine practice before the trial, a new procedure to screen for mental health problems in children was introduced in all participating CHCs. In addition to collecting outcome measures for the main trial, the intention of introducing SDQ was to facilitate the children's yearly health check-ups by providing the CHC-nurses with preschool teachers' important knowledge about the individual children.

Following the new procedure, nurses sent study information, consent forms and three sets of paper and pencil questionnaires, including SDQ (one for each of the child´s legal custodians and one for the preschool teacher), together with the standard invitation letter that parents receive about three weeks prior to the child's routine check-up. For nurses' convenience, packages including all materials were prepared, sorted by age, and delivered to the participating CHCs. Nurses then added the routine invitation letter to a package and sent it to the parents. Participating parents were asked to (a) return their completed questionnaires when attending the child's check-up at the CHC, and (b) to sign and take the teacher questionnaire to their child's preschool. Preschool teachers were instructed to complete the questionnaire and then send it to the child's CHC-nurse in the prepaid envelope provided. The nurse then reviewed the parent- and teacher SDQ during, prior to, or sometimes, after the check-up.

**Strengths and Difficulties Questionnaire.** The Strengths and Difficulties Questionnaire [14] is a well-documented instrument for measuring children's mental health, available in both parent- and teacher versions. The SDQ comprises 25 items on a 3-point scale from 0 (not true) to 2 (certainly true). It has five subscales: Emotional symptoms, Conduct problems, Peer problems, Hyperactivity/Inattention and Prosocial behaviour. A confirmatory factor analysis, conducted within the trial, showed good fit for preschool teachers, mother and fathers [15]. No cut-offs or instructions for score calculations were presented to the nurses since the SDQs only served as a discussion document at the CHC visit. Instead, a bicoloured score sheet (transparent overlay), indicating items with high scores within two different problem areas (emotional and behaviour problems), was provided for the nurses' convenience to identify possible areas of concern. Besides the five subscales, parents completed the impact supplement of the SDQ [16] which includes eight items. The first item explicitly asks whether the informant believes the child has problems (perceived difficulties). A positive answer leads to further enquiries about problems' chronicity, overall distress, social impairment and burden. Teachers only filled in the first item in the SDQ supplement. Parents were informed to complete the study questionnaires independently. Nurses were informed that children perceived as being symptomatic should be addressed in accordance with standard practice i.e. followed up at the CHC or referred to specialists (e.g. speech and language therapists, psychologists and physicians).

The study questionnaires sent to parents included SDQ but also items regarding the child's physical health, language and healthcare consumption. Furthermore, parents provided demographics and rated their own health using the General Health Questionnaire (GHQ-12) [17]. Mothers and fathers were also asked to answer items relating to their parenting practices. The Parenting Scale [18] is a 30 items questionnaire that measures dysfunctional discipline styles in caregivers. Parents are asked to indicate how they would respond to various problem behaviours by choosing between an effective and ineffective response on a 7-point scale. In the Swedish version, only two of the original three subscales were retained [19]: laxness (11 items) and overreactivity (10 items). This process evaluation focuses primarily on the use of the SDQ, as this was what the nurses actually used in their own work (along with items on child's health and development). The rest of the study questionnaire and a consent form, were collected for research purposes only. A detailed description of the Children and Parents in Focus trial (main trial) and the outcome measures collected in the trial was published in 2013 [13].

## Data collection

Materials included monitoring data, trial data, research group documents, study materials, survey data and group interviews (Table 1).

*Survey*. The survey was developed to reflect the research aims of exploring nurses' perception of the intervention and the facilitation strategies, but also to revisit the findings from the interview study, conducted in 2014 [4] (see Table 2 for items). The survey was distributed in January 2018, and 91% of the nurses involved in the trial at the time (52/57) responded anonymously.

*Group interviews*. Information on nurses' experiences of the intervention was obtained, during short (3–11 min) interviews, in groups for the nurses' convenience, in connection with educational outreach visits at the local CHCs 23–24 months into the trial. Six CHCs (private and public) were purposively sampled in order to include units of different sizes, representing rural as well as urban areas and areas with varying socio-economic conditions. The CHC were also selected to represent CHCs with observed varying capability to distribute questionnaires to enrolled children (performance). All of the nurses that were approached ($n = 16$) wanted to participate in the interviews. The interviews from 2014 informed the key topics for discussion [4]: If the nurses thought that the discussions with parents had changed since the routine with SDQ was introduced, if the nurses considered the score sheet (for SDQ) to be useful, and if the nurses had experienced difficult situations that they thought were related to using the SDQ.

## Ethical considerations

Ethical clearance was granted by the Regional Ethical Review Board in Uppsala, Sweden (Dnr 2012/437). All parents were provided with written study information sheets together with the study questionnaires, and the parents or legal guardians of all children participating in the

**Table 1. Data sources used in the process evaluation.**

| Component | Subcomponent | Data source | Indicator |
|---|---|---|---|
| Implementation | Facilitation strategies and adaptations | Research group documents*, | • Number and content of newsletters |
| | | | • Number and content of educational outreach visits at the local CHCs |
| | (Implementation efforts delivered) | Study materials** | • Number and content of educational meetings with all nurses involved in the trial |
| | | | • Assistance provided to CHCs |
| | | | • Adaptations made in study questionnaires and implementation activities |
| | Dose | Monitoring data*** | • The proportion of children enrolled at each CHC receiving the study questionnaires |
| | (the proportion of the intervention delivered) | | |
| | Reach | Trial data | • Response rate: the proportion of children receiving the study questionnaires who had at least one parent or the preschool teacher SDQ returned to the CHC |
| | (the degree to which the intended audience participated in the intervention) | | • Population coverage: the proportion of enrolled children who had at least one parent or the preschool teacher SDQ returned, irrespective of whether they had received the study questionnaires |
| Mechanisms of impact | Participant responses | Survey data, Group interviews | • Nurses' ratings of facilitation strategies used (Survey–see Table 2 for items) |
| | (How nurses perceived the facilitation programme and the intervention) | | • Nurses' ratings of the intervention's characteristics (Survey–see Table 2 for items) |
| | | | • Nurses' perspectives on the intervention (Group interviews, Survey–see Table 2 for items) |

* Newsletters and field notes on educational outreach visits, educational meetings, assistance provided and adaptations made.

** Study questionnaires (the original and all updated versions) and materials for marketing.

*** Records kept by the researchers throughout the trial on number of enrolled children and distributed questionnaires at each CHC.

**Table 2. Items in the evaluation questionnaires.**

| Item | Content | Type | Motive |
|---|---|---|---|
| 1–11 | Questions about the nurse's work experience, CHC-unit, and the time allocated to the 3-, 4-, and 5-year-old child's visits in general | Fixed responses | General questions for interpretation of the results |
| 12 | 3-point item, including 11 statements measuring the nurse's agreement with a variety of statements regarding the information sharing procedure | Fixed responses | Five statements related to Diffusion of Innovations, the qualities of the innovation that determine the level of its adoption (complexity, compatibility, relative advantage, observability and trialability). Six statements related to findings in the previous interview study |
|  | 0: 'disagree', 1: 'somewhat agree', 3: 'strongly agree' |  |  |
| 13–14 | Questions about experiences of negative reactions related to the questionnaires | Fixed responses / Open-ended | Questions related to findings in the previous interview study |
| 15–17 | Questions assessing whether the nurse thought that the information sharing procedure was too taxing for nurses or parents | Fixed responses / Open-ended | Questions related to findings in the previous interview study |
| 18 | Question regarding perceived reach by parental demographics | Fixed responses | Question related to findings in the previous interview study |
| 19–20 | Questions about experiences of difficult conversations related to the questionnaires | Fixed responses / Open-ended | Questions related to findings in the previous interview study |
| 21 | 5-point item, including 3 statements assessing how important three different facilitation strategies were for the nurse's motivation to work with the questionnaires, from 0: 'not important' to 5: 'very important' | Fixed responses | Evaluation of facilitation strategies |
| 22 | One question asking if the information sharing, at the end of the trial, had become an integrated part of the nurse's daily work | Fixed responses | General question related to perceived level of adoption |
| 23–24 | Items on 'what could have been done better' and 'Other comments' | Open-ended | General questions |

main trial provided written informed consent on behalf of their children. Completed parental forms for children without written informed consent for research served as basis for the CHC-assessment, and were sent to the research group where they were anonymised and registered for response rate counts. All nurses participating in the group interviews gave oral consent for participation and audio-recording after being provided with pertinent information about the study. The survey conducted at the end of the main trial was anonymous. The adult individuals and the parents of the minor child pictured in Fig 3 have given written informed consent (as outlined in PLOS consent form) for their photograph to be published.

## Data analyses

Dose and reach were calculated for each year of the intervention period (see Table 1 for definitions). Pearson chi-square tests were performed to test the significance of the changes in dose and reach between the study years. The p-value was set at 0.001, considering the large sample sizes.

Quantitative survey data were entered into SPSS [20] and descriptive statistics were used. Verbatim transcribed group interview data and free-text answers (survey questionnaire) were analysed using qualitative content analysis, as described by Elo and Kyngäs [21].

**Rigour.** Credibility was achieved by using interview guides and by complementing qualitative data with survey questions. To enhance *dependability* the principles of qualitative content analysis [21] were carefully followed. Furthermore, three analysts were involved in the analysis process. To reduce the risk of the respondents being afraid of expressing their true opinions about the SDQ procedure, nurses responded anonymously to the survey questionnaires collected after the end of the trial. *Transferability* was enhanced by purposive sampling of interview subjects and collecting anonymous survey data from the total eligible sample.

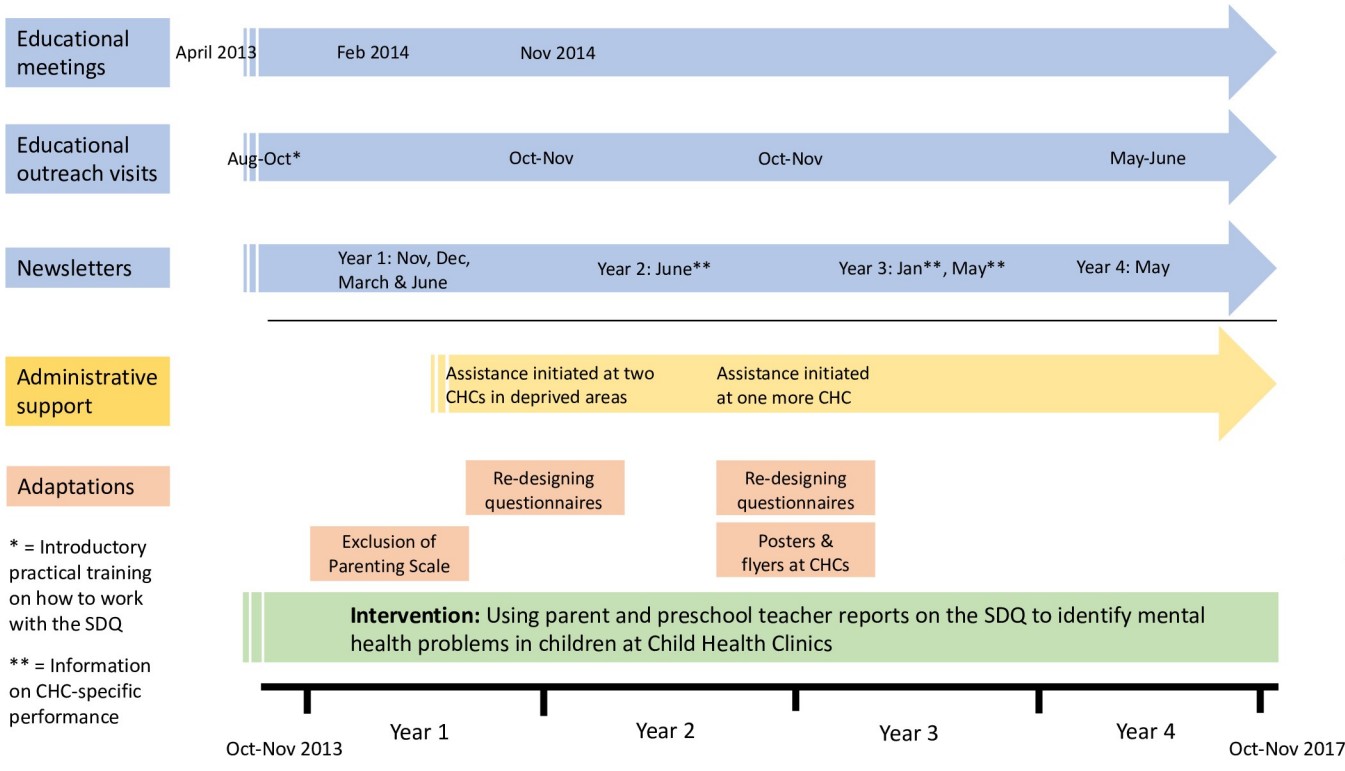

**Fig 1. Overview of the different facilitation strategies used during the intervention period.**

## Results

### Implementation

**Educational meetings and educational outreach visits.** General information about the new routine was provided at the regular educational meetings with all nurses involved in the trial (compulsory attendance).

Before the launch of the study, nurses at all participating CHCs received introductory training on how to work with the SDQ, i.e., how to distribute and collect the study questionnaires and use the score sheet (transparent overlay) indicating possible areas of concern. The training was provided via one-hour educational outreach visits from one or two members of the research team. In addition to reviewing the parent and teacher SDQ assessments, nurses were instructed to directly ask parents about their child's behaviour and then weigh their clinical observations together with the parents' description of their child. Additional educational outreach visits with nurses were held during the course of the trial. The meetings were led by external facilitators (nurses and members of the research team) and took place at the local CHCs during lunchtime for the nurses' convenience. The facilitators were instructed to ensure that all CHC-nurses working at the participating CHCs were adequately informed about the information sharing procedure, and worked with the SDQ procedure as intended. Nurses were invited to a salad luncheon and were provided with up-to-date information regarding the trial and the study questionnaires, specifically. The purpose of the meetings was to discuss nurses' experiences and thoughts about the new routine, to detect implementation barriers that needed to be addressed, and to foster a positive relationship between the researchers and the nurses.

**Newsletters.** In total, nine newsletters were sent to all individual nurses at the CHCs. In six out of nine newsletters, nurses were simply given information about the study in general and provided with feedback regarding all CHC-nurses' joint performance regarding reach of the SDQ forms. The other newsletters (nos. 5,6,7) provided information on CHC-specific performance, and how this compared to other CHCs' performance, based on dose and reach. Commercial gift cards were attached to the letters providing information on performance. However, in two letters (nos. 6,7), nurses were also informed that the CHCs with the highest performance were going to be rewarded with additional gifts for the whole CHC (worth approximately 550 USD).

In order to deal with major implementation challenges identified during the first and second year of the trial (described below), two additional strategies were included in the facilitation programme:

1. providing facilitative administrative support

2. adaptations made in materials and implementation activities when required

Fig 1 provides an overview of the facilitation strategies used during the intervention period.

**Facilitative administrative support.** Base year analyses showed that participating parents were more likely to be cohabiting, well-educated and born in Sweden compared to the municipality average. Moreover, monitoring data on CHC-specific performance indicated that some CHC's performances were low. Hence, one year into the trial, two CHCs, in deprived areas with a high rate of foreign-born parents, were provided assistance in sending out the invitation letters. Two years into the trial, the research team decided to initiate assistance at yet another CHC located in an area with similar socio-economic (SES) characteristics. Every second week, a research assistant went to the CHCs to send invitation letters and study questionnaires in the primary language used by each family, as indicated by the nurses.

**Adaptations.** Initially, two subscales of a widely used instrument measuring parenting practices, the Parenting Scale (PS) [22], were included in the study questionnaire. However, within weeks of the first-year data collection, the research team received several calls and e-mails from nurses and parents expressing discontent with the overreactivity subscale items in the PS concerning how parents handled conflicts with their children, including items about shouting at, shaking and hitting children. As a result, some parents had decided against participation in the project either because the subscale items offended them or they were worried about how the information might be used in child health centres. In an earlier intervention study in Sweden, the PS had been successfully used with parents of preschool children [19, 23]. However, the completed forms were handled by researchers without the involvement of CHC-nurses. Nurses participating in the current study reported negative reactions:

*It was worst the first year (. . .) they tore out the pages or crossed it out and just threw it away. Yes, was very annoying. (Group interview, CHC no 1, private)*

Since 1979, Swedish law has prohibited any form of corporal punishment of children, and nurses are obliged to report if they suspect a child is at risk for abuse or neglect. Thus, although unexpected, it became clear that CHCs are not a suitable setting for using the PS. Therefore, after careful consideration, the PS was removed from the set of questionnaires collected from all parents at CHCs because confidence in the study was a higher priority.

Interviews conducted in 2014 [4] showed that parents perceived the first version of the study questionnaires as difficult to follow, and also that many parents missed the instructions about leaving one of the questionnaires to the preschool teacher. To deal with these challenges, the study questionnaires were redesigned to make it easier for parents to comply with the

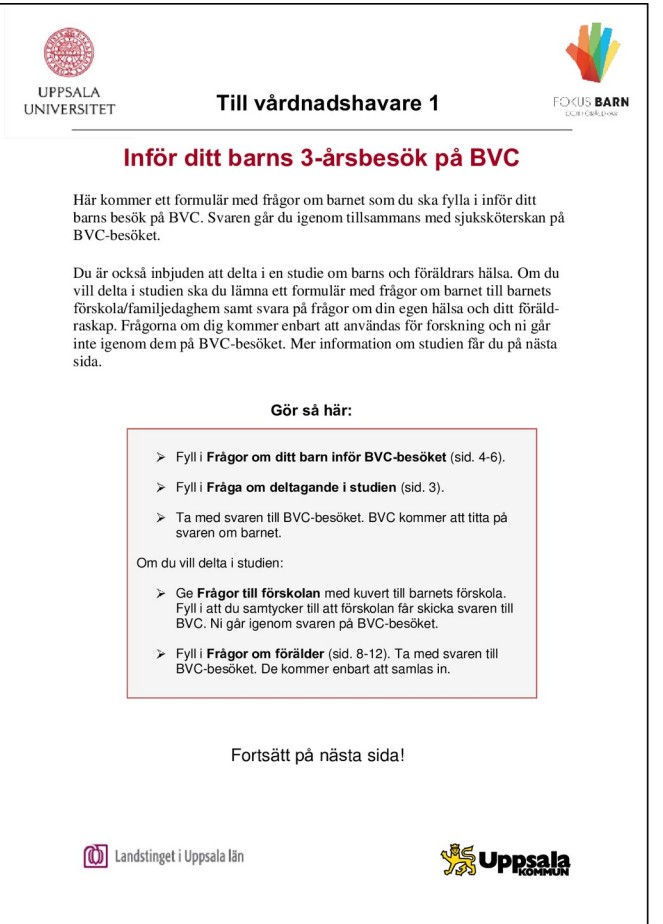
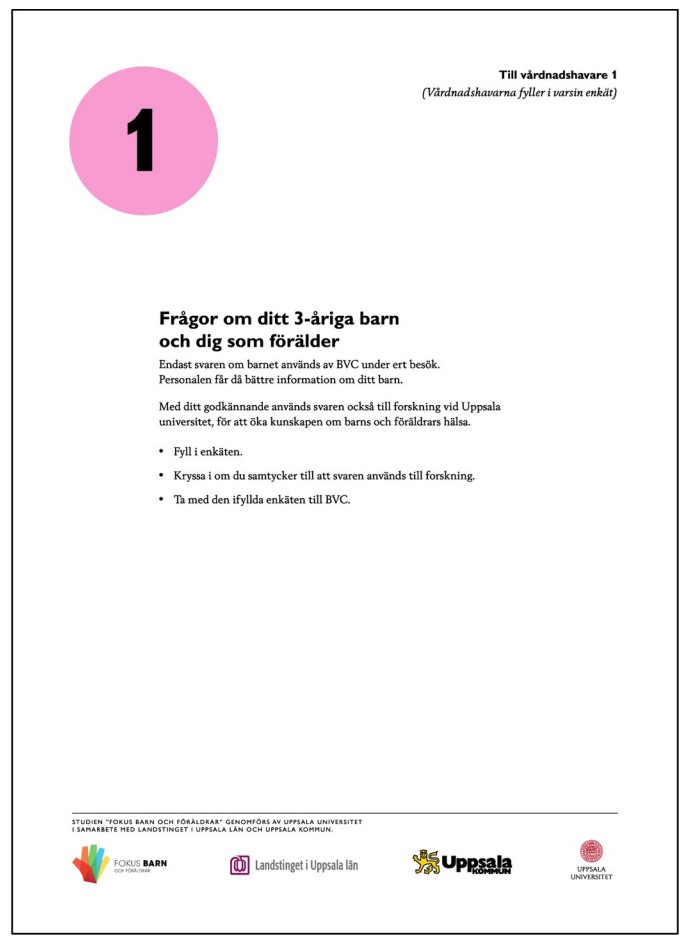

**Fig 2. The study questionnaire before and after redesign.**

desired procedure. An independent advertisement agency was employed to adapt the layout of the study questionnaires at two different time points (Fig 2). The most substantial changes in the layout were made between year two and year three. Nurses indicated that they experienced fewer negative reactions from parents after the layout redesign.

> *But, it has become better over the years as the questionnaires were redesigned. (Group interview, CHC no 1, private)*

*Increasing implementation activities.* Two years into the trial, measures to market the information sharing procedure were put into place. The advertisement agency created one poster and one flyer to communicate benefits with the information sharing (Fig 3). The poster was put up at all participating CHCs and preschools, and the flyer was placed in the CHC waiting rooms. In the fourth year, the flyer was also distributed to all participating preschools.

## Implementation outcomes

An overview of the implementation outcomes is presented in Table 3. Intervention dose and reach varied greatly between CHCs.

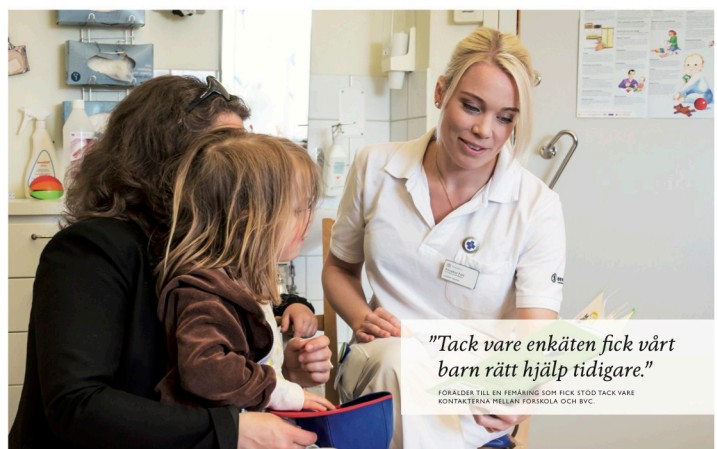

Har du fått
BVC-enkäterna?

Du som har barn i åldern 3–5 får ett brev inför ert nästa besök
på BVC. Brevet innehåller enkäter med frågor om barns och
föräldrars hälsa. För att det ska bli lite lättare för dig samarbetar
vi på landsting och universitet samt samlar allt i ett paket.

**Därför är BVC-enkäterna viktiga**

- BVC får bättre information om ditt barn inför besöket.
- Du bidrar till ökad kunskap om barns och föräldrars
  hälsa – till hjälp för alla i samhället.

**Enkäterna finns även på andra språk.**

The questionnaires are also
available in other languages.

Warsashooiyinkan waxa lagu
helayaa xitaa luqaddo kale.

هذه الاستمارات متوفرة أيضاً بلغات أخرى.

www.kbh.uu.se/
bvc-enkaterna

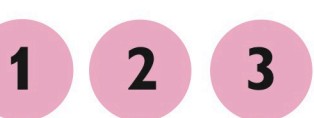

STUDIEN "FOKUS BARN OCH FÖRÄLDRAR" GENOMFÖRS AV UPPSALA UNIVERSITET
I SAMARBETE MED LANDSTINGET I UPPSALA LÄN OCH UPPSALA KOMMUN.

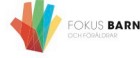 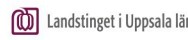 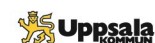 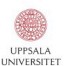

**Fig 3. Poster to communicate benefits with the SDQ procedure.**

**Dose.** The total score for dose delivered averaged 77% in year one, 84% in year two, 91% in year three and 87% in year four, indicating a high compliance with the desired practice (Table 3).

**Reach.** Response rate averaged 63% in year one, 54% in year two, 56% in year three, and 58% in year four. Population coverage averaged 48% in year one, 45% in year two, 51% in year three and 50% in year 4 (Table 3). Based on findings from a short survey conducted using a similar procedure, population coverage was expected to exceed 50%. This level was attained by 68% of the CHCs in year one, 39% of the CHCs in year two, 56% of the CHCs in year three and 53% of the CHCs in year four, indicating that the expected level remained fairly difficult to attain for some CHCs throughout the intervention period.

**Table 3. Overview of implementation outcomes by year.**

| | | | Implementation outcomes | |
|---|---|---|---|---|
| | Target population | Dose | Reach | |
| Year in study | n | % (range) | Response rate | Population coverage |
| | | | % (range) | % (range) |
| Year 1 | 6882 | 77 (28–103*) | 63 (39–87) | 48 (24–68) |
| Year 2 | 7056 | 84 (58–100) | 54 (30–71) | 45 (24–71) |
| Year 3 | 7316 | 91 (62–123*) | 56 (36–66) | 51 (22–72) |
| Year 4 | 7689 | 87 (43–107*) | 58 (36–73) | 50 (20–75) |

**Dose** is the percentage of enrolled children receiving study questionnaires (the proportion of the intervention delivered). **Response rate** is the proportion of children receiving the study questionnaires who had at least one parent or preschool teacher SDQ returned to the CHC. **Population coverage** is the proportion of enrolled children who had at least one parent or preschool teacher SDQ returned, irrespective of whether they had received the study questionnaires.

* = Percentage over 100% due to inevitable uncertainty in the distribution of enrolled children at the different CHCs during each study year. The uncertainty is related to children moving in to or out of Uppsala, parents deciding to register their child with another CHC within Uppsala, or the opening of new CHCs / closing of existing CHCs. However, data related to a child are only represented once per study year.

Measures of dose and reach at the three CHCs receiving facilitative administrative support are presented in Table 4. The level of dose improved by 15–31% for all CHCs between year one and year four. However, while the year four dose level exceeded 90% for two of the CHCs, one CHC (CHC 2) scored considerably lower, at 43%. A conspicuously low level of population coverage also reflected the low dose level.

As displayed in Table 5, both dose and population coverage increased significantly each year, from year one to year three. From year three to year four, while dose decreased significantly, population coverage remained about the same. Response rate decreased significantly only from year one to year two and remained about the same over the following years.

**Table 4. Percentage of dose delivered and of children participating in the focus intervention at the three CHCs receiving extra support.**

| | | Year 1 | Year 2 | Year 3 | Year 4 |
|---|---|---|---|---|---|
| | | (%) | (%) | (%) | (%) |
| **CHC 1** | *Enrolled children* | *n = 713* | *n = 755* | *n = 670* | *n = 698* |
| | Dose | 71 | 77 | 84 | 91 |
| | Response rate | 70 | 52 | 56 | 57 |
| | Population coverage | 50 | 40 | 47 | 52 |
| **CHC 2** | *Enrolled children* | *n = 868* | *n = 885* | *n = 865* | *n = 883* |
| | Dose | 28 | 58 | 62 | 43 |
| | Response rate | 87 | 45 | 36 | 47 |
| | Population coverage | 24 | 26 | 22 | 20 |
| **CHC 3** | *Enrolled children* | *n = 406* | *n = 450* | *n = 472* | *n = 566* |
| | Dose | 63 | 81 | 78 | 94 |
| | Response rate | 39 | 30 | 61 | 43 |
| | Population coverage | 24 | 24 | 47 | 41 |

**Dose** is the percentage of enrolled children receiving study questionnaires (the proportion of the intervention delivered). **Response rate** is the proportion of children receiving the study questionnaires who had at least one parent or preschool teacher SDQ returned to the CHC. **Population coverage** is the proportion of enrolled children who had at least one parent or preschool teacher SDQ returned, irrespective of whether they had received the study questionnaires.

**Table 5. Changes in dose and reach over study duration.**

|  | Year 1 | Year 2 | Year 3 | Year 4 |
|---|---|---|---|---|
| **Numbers ($n$)** |  |  |  |  |
| Enrolled children | 6882 | 7056 | 7316 | 7689 |
| Received the study questionnaire | 5281 | 5895 | 6628 | 6708 |
| At least one SDQ returned | 3314 | 3197 | 3725 | 3880 |
| **Percentages (%)** |  |  |  |  |
| Dose | 77 | 84 | 91 | 87 |
| Response rate | 63 | 54 | 56 | 58 |
| Population coverage | 48 | 45 | 51 | 50 |
| **Change compared to previous year ($x^2$)** |  |  |  |  |
| Dose |  | 101.7* | 159.3* | 42.7* |
| Response rate |  | 83.2* | 4.9 | 3.7 |
| Population coverage |  | 11.3* | 45.2* | 0.3 |

**Dose** is the percentage of enrolled children receiving study questionnaires (the proportion of the intervention delivered). **Response rate** is the proportion of children receiving the study questionnaires who had at least one parent or preschool teacher SDQ returned to the CHC. **Population coverage** is the proportion of enrolled children who had at least one parent or preschool teacher SDQ returned, irrespective of whether they had received the study questionnaires * $p < 0.001$

## Mechanisms of impact–Nurses' responses to and interactions with the facilitation programme and the intervention

**Nurses appreciated the facilitation strategies.** In the survey, educational meetings and educational outreach visits received "very important" scorings, at 41% and 33%, respectively. Interestingly, newsletters were considered very important by 20% and not at all important by 8% of the respondents (Table 6).

**The intervention provides a basis for better overview of the child's health and well-being.** Content analysis of the data confirmed findings from the previous interview study conducted in 2014 [4]. Specifically, nurses perceived the new routine as contributing to improved quality of the CHC check-ups by making them more structured and with increased focus on the child's mental health.

*You become a bit more specific in your question to the parent. . . that, ok, you think that he always seems to be easily disturbed and nervous? Then you can focus and go into more detail in the conversation. (Group interview, CHC no 3, private)*

Also in line with the previous findings, nurses experienced that the SDQ made parents reflect on their children's situation.

*If they have this questionnaire, then maybe they (the parents) think a little more about these things than they might have if they only came here (without the questionnaire). (Group interview, CHC no 1, private)*

Survey data also confirmed that nurses acknowledged benefits of the information sharing procedure using the SDQ (Table 7). All nurses agreed somewhat (21%) or totally (79%) with the statement that the intervention using SDQ was a good method to identify children with

**Table 6. Nurses' opinions on facilitation strategies used during the intervention period to optimise fidelity.**

| Facilitation strategy | Nurses' ratings of facilitation strategies (n = 52) | | | |
|---|---|---|---|---|
| | Not important* | Important | Very important | No opinion |
| | % (n) | % (n) | % (n) | % (n) |
| Educational meetings | - | 43 (21) | 41 (20) | 16 (8) |
| Educational outreach visits | 2 (1) | 49 (24) | 33 (16) | 16 (8) |
| Newsletters | 8 (4) | 55 (27) | 20 (10) | 16 (8) |

The survey was conducted in January 2018 (after the end of the trial). Ninety-one per cent of the nurses involved in the trial at the time (52/57) participated. Nurses were asked to rate the strategies decided upon before the launch of the trial.

* The response alternatives 'Not important' and 'Slightly important' combined.

mental health problems. However, 10% considered the method as being too burdensome (for nurses).

Almost all of the respondents experienced that the SDQ was important for their assessment of the child, and found that the method increased their knowledge of the child's mental health.

**The preschool assessment.** Seventy-six per cent of the nurses agreed totally with the statement that the preschool SDQ was important when evaluating the child's psychosocial health (Table 7).

**Table 7. Perceptions of the Strengths and Difficulties Questionnaire among Swedish nurses.**

| Item | Proportion of respondents (n = 52) | | |
|---|---|---|---|
| | Strongly agree | Somewhat agree | Disagree |
| | % (n) | % (n) | % (n) |
| Before the introduction of the SDQ, visits for 3- to 5-year-olds did not always reveal enough information about the child's behaviour and mental health | 30 (11) | 67 (24) | 3 (1) |
| SDQ is important for my assessment of the child's general health and well-being | 44 (22) | 54 (27) | 2 (1) |
| The SDQ reveals information about the child that probably would have been missed without using the form | 52 (27) | 46 (24) | 2 (1) |
| I believe that the SDQ gets parents to reflect on their children | 73 (38) | 27 (14) | 0 (0) |
| I think that the information sharing procedure using the SDQ is a good method to identify young children with mental health problems | 79 (41) | 21 (11) | 0 (0) |
| I think the preschool teacher SDQ report is important when assessing the child's mental health | 76 (37) | 24 (12) | 0 (0) |
| **Item** | | | **% (n)** |
| Do you consider the information sharing procedure to be too taxing? | | | |
| a). For parents? | Yes | | 14 (7) |
| | No | | 74 (38) |
| | No opinion | | 12 (6) |
| b). For nurses? | Yes | | 10 (5) |
| | No | | 84 (43) |
| | No opinion | | 6 (3) |
| Have you experienced any negative reactions since the introduction of the information sharing procedure? | No | | 27 (14) |
| | Yes, from parents | | 58 (30) |
| | Yes, from preschool | | 52 (27) |
| | Yes, from colleagues | | 2 (1) |
| | Yes, from the manager | | 0 (0) |
| Have you experienced difficult situations, which you think were related to the SDQ? | Yes | | 53 (27) |
| | No | | 47 (24) |
| Do you think that the information sharing procedure had become an integrated part of your daily work at the end of the trial? | Yes | | 96 (47) |
| | No | | 2 (4) |

Involving preschool teachers in the behavioural assessment of the children was considered both a strength and weakness of the intervention. Nurses noted that while they found it very useful to be provided with completed SDQs from both parents and teachers, preschool assessments were often lacking at the time of the check-up. Therefore, nurses saw a need for modification of the information sharing procedure.

*Actually, I think it would be better if they (the preschool teachers) gave (the questionnaires) to the parents (instead of directly sending it to CHC via post). (Group interview, CHC no 1, private)*

More than half of the nurses (52%) reported having experienced negative reactions related to the information transfer from preschools. In addition, some preschool teachers were still unwilling to assess children using the SDQ several years into the project (Table 7).

*Sometimes, the preschool doesn't want to do assessments of the children. They (the parents) have experienced that. (Survey questionnaire, Nurse 29)*

Finally, some nurses also noted that parents were sometimes concerned about teachers' time constraints.

*Some (parents) don't want to leave (the questionnaires) at the preschool; they say that the preschool has so much to do and don't want to give them even more. (Survey questionnaire, Nurse 26)*

**Using the SDQ in clinical practice.** Nurses highly valued the SDQs completed by parents and teachers and invested great effort in attempting to adequately use the information. They reported that they used the score sheet quite regularly in practice, as it helped them to identify possible areas of concern, particularly before they became familiar with the questionnaire. When asked whether the intervention was an integral part of the day-to-day routines of CHC at the end of the trial, 96% (n = 47) of the respondents agreed (Table 7).

Nurses' perceptions of the intervention, in relation to the five qualities that according to the Diffusion of Innovations Theory determine the level of its adoption, indicated that the information sharing suited nurses' needs. In fact, 96–100% of the nurses agreed totally, or to a certain degree, with the statements about the qualities of the intervention thought to promote its implementation (Table 8).

## Mechanisms of impact–Unexpected consequences

**Parent reactions.** More than half (58%) of the nurses reported of encountering negative reactions from parents in relation to the SDQ (Table 7). Difficulties related to the SDQ were also described in the group interviews relating to situations when parents and teachers' responses diverged.

*I had a child for whom the preschool had answered completely... (...). It didn't fit at all with the parent's picture of the child, so it became a bit... It became difficult for the mother. She felt bad about it all, because she didn't know anything. (Group interview, CHC no 3, private)*

Some nurses also reported experiences of difficult conversations where the SDQ indicated difficulties, but the parents were not concerned about their child's behaviour, and thus did not agree to any further evaluations.

**Table 8. Nurses' opinions of the intervention's qualities.**

| Characteristic | Item | Proportion of respondents (n = 52) | | |
|---|---|---|---|---|
| | | Strongly agree | Somewhat agree | Disagree |
| | | % (n) | % (n) | % (n) |
| Relative advantage | The information sharing procedure using the SDQ is beneficial compared to the standard procedure | 60 (27) | 36 (16) | 4 (2) |
| Compatibility | Collecting information from both parents and preschool teachers is compatible with the CHCs' current needs | 81 (39) | 17 (8) | 2 (1) |
| Complexity | I find the method for information sharing easy to work with | 67 (35) | 33 (17) | 0 (0) |
| Trialability | I can adapt the use of the SDQ forms to fit my own way of working | 65 (34) | 35 (18) | 0 (0) |
| Observability | The advantages of the information sharing procedure are apparent to me | 85 (44) | 15 (8) | 0 (0) |

Nurses also reported negative reactions from parents considering SDQ items to be inadequate. In those cases, parents argued that SDQ items were not age-appropriate or difficult to interpret.

In the survey, 14% of nurses agreed that the method of information sharing was too taxing for parents. In the interviews, nurses pointed out that assessing children using the SDQ can be difficult for parents, especially those with another mother tongue, resulting in them not participating in the intervention.

*Parents think there is a lot to fill in and have often forgotten to fill it in and bring it with them.* (Survey questionnaire, Nurse 26)

Almost all the nurses reported that the SDQ formed a good basis for discussion with the parents and helped them to approach sensitive topics more easily. However, one nurse indicated that she found it difficult to discuss the parents and teachers' SDQ assessments at the CHC-visit.

*I can think that these things can be hard to bring up with the parents if the child is present because they actually understand everything we are talking about. (Group interview, CHC no 6, public CHC)*

## Discussion

A mixed-methods approach was used in this process evaluation to describe and evaluate a facilitation programme developed to support the introduction of a method to assess mental health of 3–5-year old children within routine Child Health Care. Overall, process outcomes indicated that nurses were in favour of the new practice, appreciated the facilitation strategies used, and delivered the intervention as intended (dose averaged 77–91% during the intervention period). Reach outcomes, on the other hand, remained lower than expected throughout the intervention period.

The selected strategies are supported by research, showing that educational meetings and educational outreach visits can have a small effect on professional health care practice [24, 25], and that audit and feedback generally have a limited, but potentially important effect in terms of improving healthcare practice [26]. Furthermore, although there is no convincing evidence that using a combination of facilitation strategies is more effective than using single strategies [27], multi-faceted approaches is commonly viewed as more effective to change healthcare professionals' behaviour.

As part of the facilitation programme, nurses received three newsletters (nos 5, 6 and 7) providing information on CHC-specific performance from the external facilitators. The nurses were also given verbal feedback during the educational outreach visits. The audit and feedback provided was appreciated by the nurses. This is in line with Ivers et al.'s [26] conclusion that it is ideal to provide audit and feedback more than once, both verbally and in writing. Furthermore, a previous study on the role and function of facilitation [28] suggests that in order to function effectively, the facilitator's role and skills have to be adapted to the needs of the situation. In this trial, the facilitators' role contained both task-oriented and holistic elements: the facilitators were focused on providing practical support, and used educational meetings and educational outreach visits to enable nurses to reflect on and review their way of working. We believe that while the facilitators' task-oriented approach was essential to put the new SDQ procedure into practice and convince the already strained CHC-nurses to accept the new routine, the holistic approach might have been more effective in terms of sustainability. The facilitators were also easily accessible and strived to be receptive to, and act upon, problems experienced at the different CHCs. In addition, the facilitators' knowledge about the SDQ, combined with their professional background and their experience of working in the primary care context, might have made them particularly effective in their facilitating role. This assumption is supported by Harvey and colleagues [29] stating that a skilled facilitator needs to have adequate knowledge about *what* is intended to be implemented (the innovation), *who* with (the recipients) and *where* (the context).

## Ongoing support is needed if implementation challenges prevail

The importance of monitoring implementation has been reported in a review by Durlak and DuPre [30]. In the present study, yearly dose and reach measures provided a comparative overview of the extent to which the SDQ procedure was implemented over time and across different CHCs. When monitoring the performance of the CHCs, we took into account that certain CHCs were known to be struggling with economy and high personnel turnover, possibly affecting implementation, and that the questionnaires were not available in all languages, thus, excluding a certain percentage of families by default.

Regarding dose delivered, the total level increased continuously between year one and year three, but dropped 4% between year three and four. It is possible that organisational factors reduced the effect of the facilitation efforts. For example, 10% of the nurses participating in the survey conducted at the end of the trial stated that they had worked within the CHS less than one year. Furthermore, in 2015, a new national programme for child health care was introduced, affecting the 3-year visit. Finally, the three CHCs receiving facilitative administrative support were located in deprived areas with a high rate of foreign-born parents. Thus, lower response rates at those CHCs were not unexpected.

Reach levels remained lower than expected throughout the intervention period. Population coverage stayed under or just in line with 50% and response rate ended below the level (68.3%) reported in a similar intervention in Finland [31], in all study years. This result may appear disappointing, but it may neither be an indication of failure of the implementation intervention nor of using SDQ in the child health services. It is, for example, likely that implementation outcomes would have been better if SDQ had been introduced as clinical routine for all families, without asking parents to consent to research. The biggest change in population coverage was found between years 2 and 3. We believe that our efforts to make the questionnaire design more user-friendly through professional graphic design and marketing played an important role in these outcomes.

## Limitations

Results from the facilitation program might not be applicable to lower SES populations served by health units. Participating parents in the trial had higher education level, were more likely to have been born in Sweden and more likely to be cohabiting than the population average ($p < 0.001$ for all) [32]. In fact, at the CHCs with more disadvantaged populations, implementation was more difficult despite extra support, and only one of three low SES centres achieved the 50% level for population coverage.

The group interview sessions were very brief, as they had to be fitted into the busy schedules of the nurses. Thus, there was no room for in-depth exploration of nurses' views. On the other hand, this was a way to collect information from a large number of nurses.

This process evaluation used a non-experimental approach, and therefore no conclusions can be drawn on causality [7]. The aim was not to compare different models of facilitation; rather, it was to increase understanding of how complex interventions can be implemented within routine care using facilitation, and to give insight about intended adopters' experiences of commonly used facilitation strategies. On the other hand, we employed a total population approach providing high ecological validity for our findings in this process evaluation.

## Conclusions

Our findings suggest that systematically executed facilitation strategies are useful in supporting implementation in routine care and that they need to be maintained if implementation challenges prevail or new ones arise.

## Acknowledgments

We wish to thank the nurses at the participating Child Health Clinics and all parents who participated in the Children and Parents in Focus trial. We would also like to thank our colleagues Antónia Tökés, Jenny Pitkänen and Anna Backman for their help with data collection.

## Author Contributions

**Conceptualization:** Raziye Salari, Helena Fabian, Anna Sarkadi.

**Data curation:** Raziye Salari, Helena Fabian, Anna Sarkadi.

**Formal analysis:** Elisabet Fält, Raziye Salari, Helena Fabian, Anna Sarkadi.

**Funding acquisition:** Elisabet Fält, Anna Sarkadi.

**Investigation:** Elisabet Fält, Raziye Salari, Helena Fabian, Anna Sarkadi.

**Methodology:** Elisabet Fält, Raziye Salari, Helena Fabian, Anna Sarkadi.

**Project administration:** Elisabet Fält, Raziye Salari, Helena Fabian.

**Resources:** Anna Sarkadi.

**Supervision:** Raziye Salari, Helena Fabian, Anna Sarkadi.

**Validation:** Elisabet Fält, Raziye Salari, Helena Fabian, Anna Sarkadi.

**Visualization:** Elisabet Fält, Raziye Salari, Helena Fabian, Anna Sarkadi.

**Writing – original draft:** Elisabet Fält.

**Writing – review & editing:** Raziye Salari, Helena Fabian, Anna Sarkadi.

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
