## [Decision Letter · Decision Letter 0]

3 Jan 2020

PONE-D-19-20194

Facilitating implementation of an evidence-based method to assess the mental health of 3–5-year-old children at Child Health Clinics: a mixed-methods process evaluation

PLOS ONE

Dear Mrs Fält,

Thank you for submitting your manuscript to PLOS ONE. After careful consideration, we feel that it has merit but does not fully meet PLOS ONE’s publication criteria as it currently stands. Therefore, we invite you to submit a revised version of the manuscript that addresses the points raised during the review process.

We would appreciate receiving your revised manuscript by Feb 17 2020 11:59PM. To enhance the reproducibility of your results, we recommend that if applicable you deposit your laboratory protocols in protocols.io, where a protocol can be assigned its own identifier (DOI) such that it can be cited independently in the future. For instructions see: http://journals.plos.org/plosone/s/submission-guidelines#loc-laboratory-protocols

We look forward to receiving your revised manuscript.

Kind regards,

Wen-Jun Tu

Academic Editor

PLOS ONE

Journal Requirements:

2. We note that Figure 3 includes an image of a [patient / participant / in the study]. 

4. We noted in your submission details that a portion of your manuscript may have been presented or published elsewhere: This paper (or closely related research) has neither been previously published nor submitted for consideration to any other journal. However, the results presented in this paper will published in a digital comprehensive summary (Uppsala dissertations, the Faculty of Medicine).

Reviewers' comments:

Reviewer's Responses to Questions

**Comments to the Author**

1. Is the manuscript technically sound, and do the data support the conclusions?

Reviewer #1: Yes

2. Has the statistical analysis been performed appropriately and rigorously? 

Reviewer #1: Yes

3. Have the authors made all data underlying the findings in their manuscript fully available?

Reviewer #1: Yes

4. Is the manuscript presented in an intelligible fashion and written in standard English?

Reviewer #1: Yes

5. Review Comments to the Author

Reviewer #1: Facilitating implementation of an evidence-based method to assess the mental health of 3-5 year-old children at Child Health Clinics: a mixed-methods process evaluation

The authors present their journey in assessing the implementation of a mental health survey in Child Health Clinics as part of the child health evaluation. The paper offers useful insights on the implementation procedure, which will benefit others considering similar implementations. Use of mixed-methods is appropriate and useful. Pitfalls were described. The authors were responsive to difficulties and adapted their materials accordingly.

The authors opted to use the Strengths and Difficulties Questionnaire (SDQ), a well-validated instrument available in wide range of languages. Each child was assessed with the SDQ by mother, father, and teacher and Child healthcare nurses reviewed the assessments with the parents as a way to start a conversation on the child’s mental health. Therefore, this is a timely report that addresses the importance of mental health in early childhood.

The paper is well-written and clear. I would like to see, nonetheless, greater detail on a few points. Moore’s framework should be described more fully. The authors should clearly state how many children were eligible. The educational outreach needs to be described. I wonder if nurses received training to increase their understanding of typical and atypical emotional and cognitive development in the early childhood years. This would have been very helpful and a way to empower nurses. Nurses used an overlay to assess whether they needed to address issues with parents, but I think that this procedure is not sufficient. Nurses also need training to have the conversation with parents about their children’s behaviour. As the authors report, some parents, whose children screened positive, did not believe the child had a difficulty. This is a delicate conversation as it is not desirable to label children as having a difficulty when indeed they do not and equally not to discount problems that can be addressed early. Was the impact supplement of the SDQ used?

The facilitation program needs be fully described directly at the onset rather than later in the manuscript. Facilitation is an important issue because work overload and the staffs’ feelings of being over burdened is likely to be a major obstacle. How were the six Child Health Clinics selected for interviews?

The authors need to describe the Parenting Scale and exactly which subscales were used and the purpose of using them in the context of this study. Requesting too much information from parents to share with nurses is a delicate risky, because some parents may perceive it as a “nanny state.” Instead, information on parental disciplinary strategies can be brought forth in parent discussion groups at the Child Health Clinics.

It is unclear whether parents received paper and pencil questionnaires. I recommend electronic questionnaires to increase response rate for all respondents. The authors should include a description of the father responses and assess whether parents completed them independently or together. The procedure for obtaining teacher SDQ needs some modification. As the author note, teachers felt over burdened.

6. PLOS authors have the option to publish the peer review history of their article (what does this mean?). If published, this will include your full peer review and any attached files.

Reviewer #1: Yes: Alina Rodriguez

---

## [Author Response · Author response to Decision Letter 0]

1 Apr 2020

Response to Reviewers

We thank the Academic Editor and the reviewer for thorough and valuable comments on our paper. We have considered each point brought up by the reviewer, and revised our manuscript in accordance with her concerns. Below are the specific answers to the remarks. The changes are also highlighted in the “Revised Manuscript with Track Changes” file.

1.When submitting your revision, we need you to address these additional requirements.

We have reviewed the guidelines and implemented the PLOS ONE’s style requirements. 

2. We note that figure 3 includes an image of a patient / participant in the study.

Written informed consent has been obtained from the individuals imaged in figure 3, and information regarding the individuals’ consent for publication has been added to the ‘Ethical considerations’ section (page 12, line 236-238). 

“The adult individuals and the parents of the minor child pictured in Figure 3 have given written informed consent (as outlined in PLOS consent form) for their photograph to be published.”

Information about ethical and legal restrictions on sharing data publicly is explained in the revised cover letter.

3. We noted in your submission details that a portion of your manuscript may have been presented or published elsewhere: This paper (or closely related research) has neither been previously published nor submitted for consideration to any other journal. However, the results presented in this paper will published in a digital comprehensive summary (Uppsala dissertations, the Faculty of Medicine).

The reason that the previously published work does not constitute dual publication is explained in the revised cover letter.

Reviewers' Comments to Author:

Reviewer #1: 

Facilitating implementation of an evidence-based method to assess the mental health of 3-5 year-old children at Child Health Clinics: a mixed-methods process evaluation

The authors present their journey in assessing the implementation of a mental health survey in Child Health Clinics as part of the child health evaluation. The paper offers useful insights on the implementation procedure, which will benefit others considering similar implementations. Use of mixed-methods is appropriate and useful. Pitfalls were described. The authors were responsive to difficulties and adapted their materials accordingly.

The authors opted to use the Strengths and Difficulties Questionnaire (SDQ), a well-validated instrument available in wide range of languages. Each child was assessed with the SDQ by mother, father, and teacher and Child healthcare nurses reviewed the assessments with the parents as a way to start a conversation on the child’s mental health. Therefore, this is a timely report that addresses the importance of mental health in early childhood.

The paper is well-written and clear. I would like to see, nonetheless, greater detail on a few points.

Moore’s framework should be described more fully.

We agree with the reviewer regarding the need for a clearer presentation of Moore’s framework for process evaluation. Information regarding the framework has now been added to the ‘Introduction’ section (page 5, line 83-91). 

“This framework [12] argues for a structured approach to conducting a process evaluation, and recommends that such evaluation should include investigation of three key components: context, implementation and mechanisms of impact. The framework describes the components and provides a model visualizing these three functions of a process evaluation, and the relation among them. The model provides a basis for assessing and understanding factors promoting or inhibiting the embedding of complex interventions in everyday health care practice, and informs interpretation of outcomes. In this paper, we use the framework [12] to understand and evaluate the complex process involved in facilitating the implementation of a new method at Child Health Clinics.”

The authors should clearly state how many children were eligible.

Thank you for this comment. Information regarding the number of children eligible for the study, i.e. enrolled at the participating CHCs, is now added to the ‘Methods’ section (page 7, line 135-136). 

“The total number of eligible children was estimated to be 6,882 in the first year of the study and 7,056, 7,316 and 7,689 in the following years.”

The educational outreach needs to be described. 

Information about the introductory training and educational outreach provided to nurses is now added to the ‘Results’ section (page 14, line 266-277). 

“Before the launch of the study, nurses at all participating CHCs received introductory training on how to work with the SDQ, i.e., how to distribute and collect the study questionnaires and use the score sheet (transparent overlay) indicating possible areas of concern. The training was provided via one-hour educational outreach visits from one or two members of the research team. In addition to reviewing the parent and teacher SDQ assessments, nurses were instructed to directly ask parents about their child’s behaviour and then weigh their clinical observations together with the parents’ description of their child. Additional educational outreach visits with nurses were held during the course of the trial. The meetings were led by external facilitators (nurses and members of the research team) and took place at the local CHCs during lunchtime for the nurses’ convenience. The facilitators were instructed to ensure that all CHC-nurses working at the participating CHCs were adequately informed about the information sharing procedure, and worked with the SDQ procedure as intended.” 

I wonder if nurses received training to increase their understanding of typical and atypical emotional and cognitive development in the early childhood years. This would have been very helpful and a way to empower nurses. 

We agree with the reviewer’s concerns regarding the need for a deeper understanding of children’s mental health problems among nurses, and acknowledge that a successful and sustainable implementation of the information sharing procedure will probably depend on the existence of continued facilitation (including training to increase nurses’ understanding of emotional and cognitive development in the early childhood years). However, we would like to highlight that nurses have specialist training where basic child development is covered. Many nurses also have quite a long professional experience and, hence, a general feeling of the range of age-appropriate behaviours and development. 

Nurses used an overlay to assess whether they needed to address issues with parents, but I think that this procedure is not sufficient. Nurses also need training to have the conversation with parents about their children’s behaviour. As the authors report, some parents, whose children screened positive, did not believe the child had a difficulty. This is a delicate conversation as it is not desirable to label children as having a difficulty when indeed they do not and equally not to discount problems that can be addressed early.

Before the introduction of the SDQ procedure, CHC-nurses obtained information regarding the child’s behaviour by asking parents general questions. Hence, having this kind of conversations with parents has long been a part of the CHC-nurses’ tasks. However, with the new information sharing procedure using the SDQ, this information is obtained in a more structured way, potentially increasing the focus on the child’s mental health. We therefore agree that training nurses to discuss children’s behaviour with parents is highly relevant since this is a complex task, and such training should be provided (together with adequate education regarding children’s mental health) when facilitating continued use of the SDQ in routine care. To be competent in the clinical assessment of children’s behaviour, and in the conversations with parents, nurses need to have deeper knowledge about behavioural problems than being able to administer SDQ. However, it is important to note that the SDQ is intended to be used as part of the nurses’ overall clinical assessment, and not in isolation.

Was the impact supplement of the SDQ used?

Information regarding the impact supplement is now added to the ‘Methods’ section (page 8-9, line 170-174). 

“Besides the five subscales, parents completed the impact supplement of the SDQ [16] which includes eight items. The first item explicitly asks whether the informant believes the child has problems (perceived difficulties). A positive answer leads to further enquiries about problems’ chronicity, overall distress, social impairment and burden. Teachers only filled in the first item in the SDQ supplement.”

The facilitation program needs be fully described directly at the onset rather than later in the manuscript. Facilitation is an important issue because work overload and the staffs’ feelings of being over burdened is likely to be a major obstacle.

The description of the facilitation program has now been moved to the ‘method’ section (page 6-7, lines 108-123). We have also added some information about the facilitation to the ‘Discussion’ section (page 29, line 550-556).

“We believe that while the facilitators’ task-oriented approach was essential to put the new SDQ procedure into practice and convince the already strained CHC-nurses to accept the new routine, the holistic approach might have been more effective in terms of sustainability. The facilitators were also easily accessible and strived to be receptive to, and act upon, problems experienced at the different CHCs. In addition, the facilitators’ knowledge about the SDQ, combined with their professional background and their experience of working in the primary care context, might have made them particularly effective in their facilitating role. This assumption is supported by Harvey and colleagues [29] stating that a skilled facilitator needs to have adequate knowledge about what is intended to be implemented (the innovation), who with (the recipients) and where (the context).”

How were the six Child Health Clinics selected for interviews? 

The text has been clarified with additional information, which we hope provides the reader with a clearer picture of how the CHCs were selected (page 12, line 216-220). 

“Six CHCs (private and public) were purposively sampled in order to include units of different sizes, representing rural as well as urban areas and areas with varying socio-economic conditions. The CHC were also selected to represent CHCs with observed varying capability to distribute questionnaires to enrolled children (performance).”

The authors need to describe the Parenting Scale and exactly which subscales were used and the purpose of using them in the context of this study. Requesting too much information from parents to share with nurses is a delicate risky, because some parents may perceive it as a “nanny state.” Instead, information on parental disciplinary strategies can be brought forth in parent discussion groups at the Child Health Clinics.

The study questionnaires used in the main trial included both established instruments such as e.g. the SDQ and the Parenting Scale, and study specific question. A description of the Parenting Scale, and information about which subscales that were used in the study, has been added to the ‘Methods’ section (page 9, line 183-188). 

“Mothers and fathers were also asked to answer items relating to their parenting practices. The Parenting Scale [18] is a 30 items questionnaire that measures dysfunctional discipline styles in caregivers. Parents are asked to indicate how they would respond to various problem behaviours by choosing between an effective and ineffective response on a 7-point scale. In the Swedish version, only two of the original three subscales were retained [19]: laxness (11 items) and overreactivity (10 items).”

We agree with the reviewer’s concerns regarding the need to avoid that parents perceive the CHC-nurses to have a “nanny state” approach. However, the nurses were carefully instructed (verbally and in writing) not to review the parents’ answers to the items in the Parenting Scale. This was also clearly stated in the written study information sent to the parents and on the first page of the parent questionnaires. Nevertheless, it became clear that this was not enough and parents assumed the nurses might still see their answers. 

Mother and father Parenting Scale ratings were only collected as one of the secondary outcomes in the main trial. A sentence, referencing to the study protocol has been added to the ‘Methods’ section (page 9, line 191-193). 

“A detailed description of the Children and Parents in Focus trial (main trial) and the outcome measures collected in the trial was published in 2013 [13].”

It is unclear whether parents received paper and pencil questionnaires. 

A sentence has been clarified with additional detail, enabling the reader to understand that paper and pencil questionnaires were used (page 8, line 147).

“Following the new procedure, nurses sent study information, consent forms and three sets of paper and pencil questionnaires, including SDQ (one for each of the child´s legal custodians and one for the preschool teacher), together with the standard invitation letter that parents receive about three weeks prior to the child’s routine check-up.”

I recommend electronic questionnaires to increase response rate for all respondents.

The unexpectedly low response rate and population coverage presented in this process evaluation give rise to many thoughts regarding how continued use of the SDQ can be facilitated. Facilitation will, most likely, benefit from further development of the procedure in terms of modifying it to fit the users’ needs. We do agree that providing parents with an opportunity to choose between electronic questionnaires and paper and pencil questionnaires could potentially be one of the most effective actions to achieve maintenance of the information sharing procedure and should therefore be prioritized. 

The authors should include a description of the father responses and assess whether parents completed them independently or together.

Father, as well as mother and teacher SDQ scores have been presented in two published studies conducted within the Children and Parents in Focus trial; SDQ norms and cut-offs were reported by Dahlberg et al. 2020 (1), and inter-rater agreement by Fält et al. 2018 (2). The agreement analyses (Pearson and ICC) showed good agreement between parents’ ratings, with all coefficients for the subscales being .67 or higher (highest for externalising behaviours). The results indicating that mother and father responses correlates reasonably are in line with previous research on the SDQ. However, the possibility that the high level of agreement to some degree is a result of parents discussing their SDQ ratings cannot be excluded. Yet, the instructions to parents clearly stated that they should answer the items independently. Information regarding the instructions to parents has now been added to the ‘method’ section (page 9, line 174-175):

“Parents were informed to complete their questionnaires independently.”

The procedure for obtaining teacher SDQ needs some modification. As the author note, teachers felt over burdened.

The approach for obtaining teacher SDQs needs to be evaluated repeatedly and modified to fit the teachers’ needs and become more user-friendly. Future interviews with preschool teachers may result in an awareness of important modifications. A procedure using electronic questionnaires, or where preschool teachers could provide the CHC nurses with their SDQ assessments without the involvement of parents, could potentially make the procedure easier for teachers. Importantly, any approach to obtain teacher SDQs will require parental consent before the information is sent to the CHC-nurse. 

We thank the Reviewer for valuable remarks.

Literature references

1. Dahlberg A, Falt E, Ghaderi A, Sarkadi A, Salari R. Swedish norms for the Strengths and Difficulties Questionnaire for children 3-5 years rated by parents and preschool teachers. Scand J Psychol. 2020;61(2):253-61.

2. Fält, E., Wallby, T., Sarkadi, A., Salari, R. & Fabian, H. Agreement between mothers’, fathers’, and teachers’ ratings of behavioural and emotional problems in 3–5-year-old children. Plos One. 2018;13(11): e0206752

---

## [Editor Report · Decision Letter 1]

27 May 2020

Facilitating implementation of an evidence-based method to assess the mental health of 3–5-year-old children at Child Health Clinics: a mixed-methods process evaluation

PONE-D-19-20194R1

Dear Dr. Fält,

We are pleased to inform you that your manuscript has been judged scientifically suitable for publication and will be formally accepted for publication once it complies with all outstanding technical requirements.

With kind regards,

Wen-Jun Tu

Academic Editor

PLOS ONE
---

## [Editor Report · Acceptance letter]

1 Jun 2020

PONE-D-19-20194R1 

Facilitating implementation of an evidence-based method to assess the mental health of 3–5-year-old children at Child Health Clinics: a mixed-methods process evaluation 

Dear Dr. Fält:

I am pleased to inform you that your manuscript has been deemed suitable for publication in PLOS ONE. Congratulations! Your manuscript is now with our production department. 

With kind regards,

on behalf of

Dr. Wen-Jun Tu 

Academic Editor

PLOS ONE